# The Effect of the Liposomal Encapsulated Saffron Extract on the Physicochemical Properties of a Functional Ricotta Cheese

**DOI:** 10.3390/molecules27010120

**Published:** 2021-12-26

**Authors:** Zahra Siyar, Ali Motamedzadegan, Jafar Mohammadzadeh Milani, Ali Rashidinejad

**Affiliations:** 1Department of Food Science, Sari Agriculture Sciences and Natural Resources University, Khazar Abad Road, Sari P.O. Box 578, Iran; zahra_siyar@yahoo.com (Z.S.); jmilany@yahoo.com (J.M.M.); 2Riddet Institute, Massey University, Private Bag, Palmerstone North 11222, New Zealand

**Keywords:** liposomal encapsulation, saffron extract, functional ricotta cheese, physicochemical properties, texture profile

## Abstract

In this study, the encapsulation of saffron extract (SE) was examined at four various concentrations of soy lecithin (0.5%–4% *w*/*v*) and constant concentration of SE (0.25% *w*/*v*). Particle size and zeta potential of liposomes were in the range of 155.9–208.1 nm and −34.6–43.4 mV, respectively. Encapsulation efficiency was in the range of 50.73%–67.02%, with the stability of nanoliposomes in all treatments being >90%. Encapsulated SE (2% lecithin) was added to ricotta cheese at different concentrations (0%, 0.125%, 1%, and 2% *w*/*v*), and physicochemical and textural properties of the cheese were examined. Lecithin concentration significantly (*p* ≤ 0.05) affected the particle size, zeta potential, stability, and encapsulation efficiency of the manufactured liposomes. In terms of chemical composition and color of the functional cheese, the highest difference was observed between the control cheese and the cheese enriched with 2% liposomal encapsulated SE. Hardness and chewiness increased significantly (*p* ≤ 0.05) in the cheeses containing encapsulated SE compared to the control cheese. However, there was no significant difference in the case of adhesiveness, cohesiveness, and gumminess among different cheeses. Overall, based on the findings of this research, liposomal encapsulation was an efficient method for the delivery of SE in ricotta cheese as a novel functional food.

## 1. Introduction

Ricotta is an unripened cheese that is obtained by coagulation of protein through acidification with organic acids (e.g., acetic acid, lactic acid, and citric acid) and heating [1,2,3,4,5,6]. This dairy product is normally procured from whole sweet whey, a mixture of whey and whole milk, skimmed milk, or skimmed milk powder in different ratios [7]. Ricotta, which has a soft, grainy, firm texture, light sour taste [8], and a monotonic yellowish-white color, is manufactured without the addition of any starter culture [9]. This type of cheese is widely consumed worldwide and is appreciated for its flavor and nutritional benefits. The nutritional values of ricotta cheese can be improved even further by adding health-promoting bioactive components such as antioxidants, dietary fiber, and vitamins. Therefore, this dairy product is an ideal carrier for the delivery of bioactive compounds from natural sources.

Saffron is the main source of potent bioactive compounds such as crocin, picrocrocin, and safranal, which possess several health-promoting benefits. The bioactive compounds in saffron have been known effective for the treatment of certain cancers, cardiovascular and cerebrovascular illnesses, dysentery, measles, asthma, urological infections, cough, and stomach disturbances. Additionally, it possesses tranquilizing, anti-tumoral, anti-ischemic, anti-anxiolytic, anti-inflammatory, neuroprotective, DNA-protective, and especially antioxidative attributes [10,11,12,13,14,15]. For this reason, contrary to its high price, the use of saffron has increased in recent years, owing to the growing consumers’ demand for food products with added functional properties [16]. This plant has several potent bioactive (antioxidant) compounds, the most potent of which are crocin, picrocrocin, and safranal, which are responsible for the distinctive color, taste, and aroma of saffron [11,17].

While products such as saffron extract (SE) have been successfully developed and used in the pharmaceutical and nutraceutical industries, the application of SE in the food industry is limited. This is because of the instability of its bioactive compounds during processing and storage (i.e., in the existence of the factors such as temperature, pH, light, oxygen, enzyme, proteins, and metallic ions) and digestive conditions [18,19]. In this regard, encapsulation is a helpful technique for coating saffron bioactives (e.g., crocin, picrocrocin, and safranal) for not only improving their chemical stability in the food systems (during processing and storage) but also their guarded or controlled release in the gastrointestinal tract [20,21].

Several types of encapsulation methods have been studied for their use in the food industry, among which nanoliposomes (lipid-based nanocarriers) are one of the most promising systems [22]. This is due to the wide range of potential diameters and surface charges, nutrient properties, as well as their capability to encapsulate both hydrophobic and hydrophilic compounds. Other advantages of liposomes are their high encapsulation efficiency, low toxicity, and scalability at the industrial level [23,24]. Another benefit of using nanoliposomes in the food and nutraceutical industries is that they minimize the negative effect of bioactives on the sensorial characteristics of the final product [25]. In general, the mechanism of action of a liposome is to create a barrier around its contents, which is stable against not only free radicals, but also oral and gastric enzymes, alkaline solution, gastric juice, bile salts, and intestinal flora that are present in the human body, and the encapsulated compound(s)/ingredient(s) can be released in the target organ/medium [4,26]. On the other hand, liposomes have a wide range of particle sizes, from macro to nano-sized liposomes [27]. In addition, liposomes may have one or more bilayer membranes, and based on the composition of the lipids, method of preparation, and also their diameter, liposomes can be classified into several types; i.e., multilamellar vesicles, small unilamellar vesicles, large unilamellar vesicles, giant unilamellar vesicles, and multivesicular vesicles [28].

The use of encapsulated bioactive ingredients and plant extracts in cheese products has already been reported. For example, Jeong et al. [29] investigated the physicochemical, microbial, and sensorial properties of Queso Blanco cheese supplemented with powdered microcapsules of tomato extract. They found that gumminess, chewiness, and hardness increased with increasing the concentrations of microcapsules in cheese. The addition of the manufactured microcapsules significantly increased the lycopene content of the cheese, while it also improved the texture properties of Queso Blanco cheese. Niro et al. [30] manufactured a functional ricotta cheese by adding a probiotic of Lactobacillus paracasei and two prebiotics (inulin and chestnut flour) in the formulation of the Ricotta cheese. They reported that there were no significant changes in the sensory profile of the cheese after the addition of 3% inulin, while sensory attributes were negatively affected by adding the chestnut flour or when the symbiotic formulation was added. Hamdy and Hafaz [31] investigated the combined effect of dried rosemary, thyme, and basil with fresh garlic on the quality characteristics of ricotta cheese during the 21 days of storage. They reported that the addition of these ingredients did not have any significant effect on the physicochemical properties of ricotta cheese. Additionally, 0.5% thyme powder with 1% garlic enhanced the antioxidant properties of ricotta cheese, but in general, the treated cheese exhibited stronger phenolic content and antioxidant activities than the control sample. Rashidinejad et al. [4], in a series of systematic experiments, evaluated the effect of liposomal encapsulation on the recovery and antioxidant properties of green tea catechins incorporated into a hard-low-fat cheese following the in vitro simulated gastrointestinal digestion. These scientists [4] observed that the liposomal encapsulated catechins or green tea extract at a concentration of 125–1000 ppm enhanced both phenolic properties and antioxidant capacity of the cheese without affecting its chemical composition or pH. Nevertheless, to the best of our knowledge, no work has been reported on the manufacture of a functional ricotta cheese containing nanoliposomal encapsulated SE and the effect of saffron bioactive compounds on various properties of this type of cheese. Therefore, this study aimed to assess the feasibility of the manufacture of a functional ricotta cheese using liposomal encapsulated saffron extract and the effect on the physicochemical and textural properties of this cheese.

## 2. Materials and Methods

### 2.1. Materials

Saffron was sourced from a local supplier in Khorasan-e-Razavi, Mashhad, Iran. Crocin was supplied by Pooyesh Darooye Sina (Mashhad, Iran). Soy lecithin was purchased from Novintejarat Arvin (Tehran, Iran). Milk powder was supplied by Golshad Ltd. (Mashhad, Iran). Sodium acetate and acetic acid, lactic acid (98%), sulfuric acid (98%), and sodium hydroxide were obtained from Merk Company (Darmstadt, Germany). All other chemicals and reagents were from analytical grade.

### 2.2. Preparation of Saffron Extract

For the extraction of saffron bioactives, first, 10 g of saffron powder was mixed with 150 mL distilled water in an Erlenmeyer flask. Then, the Erlenmeyer flask was covered with aluminum foil to prevent exposure to light before being placed in an incubator shaker (SHE, Noor Sanat Ferdous, Iran) for 24 h at 38 °C. A high-sheer mixer (10,000 rpm for 10 min, T25, IKA, Germany) was used to maximize the extraction of saffron bioactive compounds. Finally, the extract was filtered under vacuum with Whatman No. 41 filter paper (Little Chalfont, Buckinghamshire, United Kingdom), freeze-dried (VaCo 5, Zirbus technology, Germany), and stored in a freezer (−18 °C) until further use [20].

### 2.3. Preparation of Nanoliposomes Containing Saffron Extract

To prepare nanoliposomes, first, different concentrations of soy lecithin (in ratios of 0.5%, 1%, 2%, and 4% *w*/*v*) were dispersed in in food-grade acetate buffer (pH = 3.8) containing 0.25% *w*/*v* saffron extract (SE) using a magnetic stirrer (HPMA 700, Khordad-tadjhize Sharif, Iran) for 30 min. The concentration of SE was kept constant at 0.25% *w*/*v*. The dispersion was mixed using a high-shear blender (T25, IKA, Germany) at 24,000 rpm (5 × 1 min bursts) following the method published by Rashidinejad, Birch, Sun-Waterhouse, and Everett Rashidinejad, Birch, Sun-Waterhouse and Everett [4]. Finally, to manufacture homogeneous liposomes (in terms of size), the dispersion was processed using an ultrasound probe (UP400A, Advanced Technology Development, Iran) for 2 min (10 s on, 1 s off; with a power of 100 watts and a frequency of 20 KHz).

### 2.4. Particle Size and Zeta Potential

The average particle size and zeta potential of samples were measured by a Zetasizer Nano (SZ100, Horiba, Japan). All samples were diluted in distilled water (1:200), and the measurements were carried out at 25 °C [6].

### 2.5. Physical Stability of Nanoliposomes

5 mL of each sample was taken in a falcon tube (50 mL) and centrifuged (T4-50 CC, Orum-tadjhize, Iran) at 3500 rpm for 15 min in order to disturb the dispersion and obtain a two-phase material [6]. The physical stability of nanoliposomes was calculated via the following equation:(1)$$NS\;\left(\%\right)=\left(F_{ev}/I_{ev}\right)\times 100$$
where *NS*: nanoliposome stability, $F_{ev}$: final volume of the nanoliposome, and $I_{ev}$: initial volume of the nanoliposome.

### 2.6. Encapsulation Efficiency

To determine the encapsulation efficiency (*EE*), 10 mL of nanoliposomal dispersion was centrifuged (RF 10000, Orum Tadjhiz, Iran; 4500 rpm, 4 °C, 15 min), and the supernatant was dissolved in 0.1 N chloroform to disrupt the nanoliposomal structure and release saffron bioactives. Next, the clear supernatant was filtered through a 0.45 µm-sized Millipore filter (Sanmin, Taiwan), and absorbance was read using a spectrophotometer (Alpha-1502+, Mehr Tadjhiz, Iran) at the wavelength of 440 nm [32]. The EE was calculated using Equation (2).
(2)$$EE\;\left(\%\right)=\left(A_{T}-A_{F}\right)/A_{T})$$
where *EE:* encapsulation efficiency, $A_{T}$: total amount of saffron bioactive compounds (crocin), and $A_{F}$: the amount of free saffron bioactive compounds (crocin).

The standard curve was plotted using six different concentrations of pure crocin in methanol [33]. The concentration of free crocin was evaluated using this standard curve with a linear equation (Equation (3); Figure 1).
(3)$$A_{440}=0.1375C_{c}+0.002\;\left(R^{2}\right)=0.9992$$
where $A_{440}$: absorbance at 440 nm and $C_{c}$: crocin concentration.

### 2.7. Manufacture of the Functional Ricotta Cheese

First, a certain amount of milk powder was dissolved in water (40:60), and the pH was adjusted to 7.10 with sodium hydroxide (NaOH) 1N. The mixture was then heated at 86 °C for 20 min [34] and then cooled down to 55 °C before the pH was decreased to 5.00 with lactic acid 10% (*v*/*v*). After that, liposomal encapsulated SE (containing 2% lecithin) was added to milk at different concentrations (0%, 0.125%, 1%, and 2% *w*/*v*) [29]. The curd was separated using centrifugation (4000 rpm, 10 min), followed by placing in cheesecloth for 24 h in the refrigerator to drain the corresponding whey [34]. After preparation, cheese samples were kept in plastic bags at 5 °C until used. Since ricotta is a fresh type of cheese, the analyses were carried out on the first day of manufacture.

### 2.8. Compositional Analysis of Cheese Samples

The chemical composition (moisture, protein, fat, and ash contents) of cheese samples was measured according to AOAC methods [29], and the following equations were used for the corresponding calculations:(4)$$Mc\;\left(\%\right)=(\left(W_{a}-W_{b})/W_{s}\right)$$
where Mc = moisture content, $W_{a}$ = weight of plate and sample before placing in the oven, $W_{b}$ = weight of plate and sample after placing in the oven, $W_{s}$ = weight of sample.
(5)Dry matter (%)=100−moisture content
(6)$$N\;\left(\%\right)=(\left(V_{a}\times 1.4008\times 0.1)/W_{s}\right)$$
where N = nitrogen%, $V_{a}$ = volume of consumed acid, $W_{s}$ = weight of sample.
(7)Pc (%)=N×Pf
where Pc = protein content and Pf = protein factor.
(8)$$A\left(\%\right)=(\left(W_{a}-W_{b})/W_{s}\right)$$
where A = Ash, $W_{a}$ = weight of sample and jug after placing in the oven, $W_{b}$ = weight of empty jug, $W_{s}$ = weight of sample.

### 2.9. pH

The pH value of cheese samples was determined using a calibrated pH meter with a glass electrode (PB-11-P11.1, Sartorius, Germany) [35]. For this test, 10 g of cheese was mixed in 50 mL of distilled water and then was transferred to a 100 mL volumetric flask and made up to volume with distilled water. The pH value was read after the insertion of the pH electrode and when a stable value was reached.

### 2.10. Cheese Color Analysis

Cheese color was measured by an IMG-Pardazesh colorimeter (CAM-System XI, Abzarkaran Fan-Pooyaye Shomal, Sari, Iran), based on L* (brightness), a* (red and green), and b* (yellow and blue). The device was equipped with a LED lamp and with a radiation angle of 45 degrees and power of 10 watts. A piece of cheese was taken into a plate and placed into the instrument and photographed by a digital camera with a resolution of 480 × 640 frames per second, and then the values of L*, a*, and b* were recorded [36].

### 2.11. Texture Analysis

The texture profile analysis (TPA) was evaluated by a texture analyzer (TA, Koppa Pajoohesh, Iran) with a flat glass probe (TAPCY10-50.8 mm D, 20 mm L), according to the method from Ortiz Araque, Darré, Ortiz, Massolo, and Vicente Ortiz Araque, Darré, Ortiz, Massolo and Vicente [36] with some modifications. The texture analyzer device had a 100 mm workbench width and 200 mm of guttural depth, as well as a 200 mm working height for flexibility in testing. Cheese samples were prepared cylindrical with 20 mm of diameter and 10 mm of height and were placed in the sample station of the texture analyzer apparatus. After that, the samples were compressed 50% at a speed of 1 mm/s, and in each measurement, the sample was compressed twice. The data for hardness (g), adhesiveness (mJ), cohesiveness, gumminess (g), and chewiness were obtained.

### 2.12. Statistical Analysis

Duncan’s multiple range test at the 95% level of probability and one-way analysis of variance (ANOVA) were accomplished to analyze the results, and significant differences were recorded at the level of *p* ≤ 0.05. All statistical analyses were performed using SPSS software (Version 22, IBM, Armonk, NY, USA). All samples were prepared in triplicate, and all tests for each sample were performed in triplicate. The results were reported as mean ± standard deviation.

## 3. Results and Discussion

### 3.1. Liposome Size

The particle size in a colloidal nano system such as the liposomal system performs an important role in specifying its properties [37]. By decreasing the particle size from micro to nanoscale and increasing the ratio of surface to volume, different traits of the system improve; bioavailability, water-solubility, colloidal stability, accessibility, turbidity, and encapsulation efficiency. Therefore, to decrease the particle size, a high-sheer mixing process was used in this study.

The average sizes of nanoliposomes are presented in Table 1. According to the results of this analysis, loading liposomes with SE in all concentrations of lecithin increased the average diameter of liposomes, with a significant difference (*p* ≤ 0.05) being observed among different treatments. This is consistent with the results of Gibis et al. [38], who encapsulated 0.1% grape seed extract in a similar procedure. In addition, the findings of Rashidinejad et al. [4], who encapsulated green tea catechins and green tea extract, and Alexander et al. [39], who examined the incorporation of phytosterols in soy phospholipids nanoliposomes, demonstrated that increasing the concentration of soy phospholipids caused an increase in the size of the nanoliposome particles. The increase in particle size of liposomes is a generic phenomenon of their mutability due to the association of unstable liposomes during processing and storage [4].

Another factor that can affect the particle size of the manufactured liposomes is the method of their preparation. In similar studies, polyphenols of green tea were encapsulated by different methods, such as a thin-film layer using a combination of ethanol injection and high pressure. In this way, the particle size obtained was smaller than 160 nm, while in the case of the present study, we obtained a particle size in the range of 155 nm to 208 nm. This difference probably depends on the amount of mechanical force; i.e., when stronger force is used, the particle size becomes smaller [40,41]. Another important factor is the use of organic solvents such as ethanol, as it changes the charge and degree of ester stability, which in turn may decrease the particle size [42]. The ratio of bioactive substance and phospholipid is also a crucial factor that can affect the particle size of a colloidal system such as liposomes. By reducing the ratio of nucleus to membrane or, in other words, by increasing the concentration of active substances, the size of nanoliposomes increases, which can be related to the fact that the active materials take up a large space of nanoliposomes [43,44].

Lecithin is the main source of liposomes and cholesterol is the membrane stabilizer. In liposomal membrane, which is rich in lecithin, the arrangement of acyl chains in one direction causes the reduction in spaces created by the large polar groups in the lipid head, and as a result, the interactions between the chains increase. Cholesterol keeps acyl chains tilted to one side in a straight line and filled in the gaps between them. Therefore, the placement of cholesterol in the bilayer structure leads to an increase in the arrangement density of phospholipid molecules. There have been more reports regarding the effect of cholesterol on liposomal size, concluding that the method for the preparation of liposomes seems to be more effective than others [45]. Several other factors can also affect the particle size of liposomes; e.g., the ratio of phospholipids to the active substance, the method of preparation of liposomes, temperature, and type of the membrane stabilizer (if used in the formulation) [42].

### 3.2. Zeta Potential

In this study, as shown in Table 1, increasing the concentration of lecithin significantly increased the zeta potential at all concentrations toward the negative range (*p* ≤ 0.05). This indicates the high stability of this colloidal system. Phospholipids that contain phosphate groups on the surface of the membrane increase the stability of nanoliposomes due to the ionization of phosphate groups in the aqueous system. On the other hand, in the present study, the surface potential of all samples was negative (ranging from 33.7 to 44.3 mV), owing to the presence of lecithin as an anionic emulsifier in the formulation of nanoliposomes [46]. Commonly, several factors affect the surface charge of the particles in colloidal systems such as liposomes; e.g., type and concentration of phospholipid, type and concentration of stabilizer, type and concentration of active compound, ionic strength of the environment, and temperature.

Zeta potential is an indicator of liposomes stability, so its measurement is useful in controlling the agglomeration and deposition of nanoliposomes, which are important factors in terms of physical stability [47]. Zeta potential is a subordinate of the surface charge of the lipid vesicles, absorbed layers on the surface, and essence of the environment in which the liposomes are dispersed [6]. Increasing zeta potential leads to increasing the repulsive force between the particles, and accordingly, it prevents the particles from colliding and accumulating. On the other hand, increasing the particle charge increases the interaction with target cells and thus increases the delivery of the bioactive compound(s). However, samples with low zeta potential have a slight repulsive force that results in binding the particles and causes physical instability [40]. Generally, for the particles to be resistant to electrostatic repulsion, the zeta potential of the entire colloidal system must be greater than ±30 mV. In addition, another factor that influences liposome stability is the attendance of repulsion force on the surface of the particles (decrease their accumulation), for which the value of the zeta potential must be lower or higher than −25 to +25, respectively [6]. Correspondingly, in the case of the liposomes manufactured in the current study, because of their high negative charge, strong physical stability was expected.

One factor that could affect the zeta potential of a liposomal system, although not applicable in the case of the current study, is the presence of membrane stabilizers such as cholesterol. Since cholesterol is a neutral molecule, the negative charge of the particles could be due to the formation of a hydrogen bond between the choline group in phosphatidylcholine and the hydroxyl group in the cholesterol head. This may result in the entrance of the positively charged choline group to the membrane, and phosphatidylcholine is negatively charged to the surface of the membrane. This, in turn, increases the negative charge of the particles and electrostatic repulsion between them [48]. Similar findings have also been reported in the case of gammaoryzanol, where the zeta potential and the stability of a liposomal system increased by using gammaoryzanol as a membrane stabilizer. Probably, gammaoryzanol molecules could increase the phosphate groups at the membrane surface by hydrogen bonding of its hydroxyl group with the choline group, leading to increasing the negative charge on the membrane due to the presence of sterols [49].

### 3.3. Physical Stability of Nanoliposomes

The results of the physical stability in the present investigation (Table 1) illustrate that all liposomal formulations were highly stable (almost greater than 92%) in terms of resistance to centrifuge force. There was a significant difference between samples containing 0.5% and 2% lecithin (*p* ≤ 0.05), with no significant difference between other samples. What stands out is that the physical stability decreased with increasing lecithin concentration, so that the formulations with the lowest (0.5%) and the highest (4%) lecithin content had the maximum and minimum stability, respectively. Tavakoli, Hosseini, Jafari and Katouzian [6], in a study on the nanoliposomes containing olive leaves, achieved similar results to those achieved in the present study. These researchers stated that particle size was an important factor influencing the state of ripening and consequently the instability of the nanoliposomes system. Ghorbanzade et al. [50], in their study on the liposomal encapsulation of fish oil, also observed that the centrifugal stability of the samples was 70.03% and concluded that this level of stability could be due to the presence of phospholipids in the structure of nanoliposomes.

Generally, the physical stability of liposomes is affected by the composition of the external environment, the particle size of liposomes, the number of layers, the structure of phospholipids, and liposome-producing techniques [50]. Liposomal systems tend to lessen the energy in the bilayer structure. The assimilation of the vesicles plus their conflict, which leads to the assimilation of the liposome membranes, is the major reason for the instability of such a system [51]. Most of the emulsifiers can cause instability of liposomes under certain conditions, which may be due to the critical concentration of micelles at a high concentration of lecithin [52]. In addition, in the case of the current study, where a slight instability (2%–8%) was observed, some compounds of SE (mainly glucose, gentiobiose, and picrocrocin) contain hydroxyl groups were expected to separate into the negatively charged groups of oxygen under the experimental conditions. Accordingly, these groups could form hydrogen and ionic bonds with positively charged choline, which is an important part of phospholipids in the nanoliposomes structure. This contributes to the Ostwald process, which is an important factor in physical stability of emulsions and other nanostructural dispersions [53].

### 3.4. Encapsulation Efficiency (EE)

As seen in Table 1, in the present experiment, EE increased with increasing lecithin concentration, but there were no significant differences observed among the formulations of nanoliposomes with less than 4% lecithin. EE is one of the most important parameters in the evaluation of nanocarriers, and in the case of liposomes, it depends on several factors such as the nature of the coating material (lipophilic or hydrophilic), the nature and concentration of phospholipids, method of preparation, and environmental conditions (e.g., pH and temperature). Bouarab et al. [54], in a study on the production of liposomes from soy lecithin, salmon, and docosahexaenoic acid phospholipids for the encapsulation of cinnamic acid, reported that the type of phospholipid and particle size of nanoliposome affected EE. They [54] explained that there was a straight correlation between increasing particle size and decreasing the EE in their experiment. In this regard, ref. [55] in a study on the nanoliposome containing vitamin C and coated with chitosan reported that liposomes with smaller sizes were able to encapsulate more vitamin C, possibly due to the presence of a greater number of nanoparticles in the system.

In the case of the current experiment, each bioactive compound in SE, depending on its chemical structure, could locate in either hydrophobic or hydrophilic part of the liposomal structure. Crocin is the most important/potent compound in SE, and because of its hydrophilic properties, it is expected to bind to the polar (inner) parts of the phospholipid molecules via hydrogen binding. As opposed, the lipophilic compounds were expected to locate in the lipid wall of liposomes. Accordingly, the ability of liposomes to retain hydrophilic compounds when these systems are in the aquatic or biological ambiance is less than hydrophobic compounds. This is due to the high distribution coefficient of lipid-water in hydrophilic compounds, and lipophilic active substances do not leak from the liposome structure because of the slow rate of diffusion. In a study by Gonnet et al. [56], it was found that the maximum concentration of active compounds was attributed to their polarity, which determines their situation in the bilayer membrane of liposomes. Carotenoids with more polarity, for example, accumulate in the bilayer membrane structure, so a higher concentration of these compounds attaches to the nanoliposomes.

Crocin, which is the main polar compound in the carotenoid metabolism group in saffron stigmas, has been identified in the form of esters of crocetin glucose (beta-glucopyranosyl) and gentiobiose (beta-di-glucopyranosyl di-glucose) [57]. Therefore, more molecules of this compound are located in the structure of the bilayer membrane, and fewer of them are in the center of the bilayer of the nanoliposomes membrane. This can probably be considered as a justification for the high EE obtained in this study. Our findings in this study are also consistent with those reported by Alexander et al. [39], who examined the encapsulation of phytosterol in soy phospholipids nanoliposomes. They reported that increased soy phospholipid content (100 to 250 mg/mL) resulted in increased EE of ascorbic acid (15.8% to 32.7%, respectively).

The correlation matrix between nanoliposomes tests is presented in Table 2. As shown in this table, there is a strong correlation between the particle size of nanoliposomes and all three parameters (zeta potential, encapsulation efficiency, and physical stability) so that as particle size increases, the zeta potential and physical stability decrease. On the other hand, the particle size has a direct correlation with encapsulation efficiency, and in this way, the encapsulation efficiency increases with increasing the particle size.

### 3.5. Chemical Composition and pH of Ricotta Cheese

The chemical composition and pH of ricotta cheese fortified with different concentrations (0%, 0.125%, 1%, and 2% *w*/*v*) of liposomal SE on the day of manufacture are shown in Table 3. In terms of composition, as seen in this table, there were no significant differences (*p* > 0.05) between control cheese and the cheese samples enriched with liposomal extract (except for moisture content/dry matter after 1% concentration). This indicates that the fortification of cheese with encapsulated SE did not have any negative effect on the chemical composition of ricotta cheese in this experiment. The reason for the slight increase in the moisture content of the ricotta cheese after the addition of 1% liposomal SE is in agreement with the observations of [58], who found an increase in the moisture content of Cheddar cheese after the addition of liposomes. These researchers explained that this trend could be due to the fact that liposome membranes bind water at their surface, and this water can be retained in the cheese structure accordingly [58]. Likewise, Drake et al. [59] reported that lecithin that contains both hydrophilic and hydrophobic properties could associate with both fat and moisture. Therefore, in this study, the presence of lecithin as a basic and major component in the nanoliposomes formulation may be a reason for increasing moisture in the samples enriched with the higher concentrations of nanoliposomes. However, this did not result in any significant change in the composition of ricotta cheese samples.

About pH values, as shown in Table 3, there was no significant difference (*p* > 0.05) between the control sample and cheeses containing liposomal encapsulated SE up to 1%. The highest pH was found in control cheese, and the lowest was seen for the cheese enriched with 2% liposomal extract, which is inconsistent with increased moisture in the samples containing liposomal extract. This can be explained by the fact that increasing moisture causes the production of fatty acids, as well as the conversion of lactose to acid lactic, both of which, in turn, decrease pH [60]. Another reason for the slight decrease in pH after the addition of liposomes can be the presence of acidic compounds (acetate buffer, pH = 3.8) in the formulation of nanoliposomes containing liposomal SE that has been trapped in the remaining whey. Jeong et al. [29], who studied the physicochemical properties of Queso Blanco cheese supplemented with powdered microcapsules of tomato extract, also achieved similar results in terms of the decrease in cheese pH. One of the reasons for decreasing the pH of the medium that such bioactives are added to can be the acidic pH of these compounds [29].

### 3.6. Color Analysis

Since color has a direct influence on the acceptability of food products by consumers, it is very important that the additives do not affect this parameter negatively [61]. As shown in Table 4, L* was decreased after the addition of encapsulated SE while the other two indices (i.e., a* and b*) were increased, even though these changes were not significant in the case of all concentrations of the liposomal encapsulated extract. The color parameters were not affected by the addition of low concentrations (i.e., 0.125% and 1%) of liposomal encapsulated extract, and the maximum difference of color parameters was seen between the control sample and sample enriched with 2% liposomal encapsulated extract. Generally, these changes are because nanoliposomes containing SE are lightly orange, and thus, the color parameters of ricotta cheese are affected. Such changes were also obvious to the naked eye, as seen in Figure 2.

The a* index, which is associated with the green color of extract, is mainly related to the chlorophyll pigments of saffron. Additionally, the increase in yellow or a* index in the cheeses that were supplemented with liposomal extract is due to the presence of the carotenoids in the SE; e.g., safranal, crocin, picrocrocin, crocetin, and their derivatives [62]. While the incorporation of higher concentrations of encapsulated SE in the present study resulted in significant changes in the color of ricotta cheese samples, no conclusion can be made on whether such changes can be seen as positive or negative by the consumers. A systemic acceptability test and consumer analysis, which could not be completed during the present study, will answer this question.

### 3.7. Texture Analysis

Texture profile analysis (TPA) is a beneficial index of the textural quality of a cheese product, which correlates well with sensory parameters [63]. In this study, the TPA of control ricotta cheese and the cheese fortified with different concentrations of liposomal SE were carried out for hardness, adhesive, cohesiveness, gumminess, and chewiness, the data for which are shown in Table 5. Hardness (g) is the quantity of force required for the compaction [64]; adhesiveness (mJ) of cheese is what is required for pulling a cheese sample out from a surface [65]; cohesiveness is the ratio between force and time for the areas of two pressures; gumminess (g) is the product required of cohesiveness by hardness [64]; and, chewiness is the energy required to chew a solid food until it is swallowable [65]. Generally speaking, the texture of cheese depends on the microstructure and chemical composition, especially fat, salt, and total solids content [64].

As presented in Table 4, while there was no significant difference (*p* > 0.05) among all cheeses in terms of adhesiveness, cohesiveness, or gumminess, hardness, and chewiness of the ricotta cheese samples containing liposomal SE were increased significantly. According to [29], the pH of Queso Blanco cheese could influence the texture, and as pH decreased after the addition of high concentrations of microcapsules containing lycopene extract, hardness, gumminess, and chewiness were increased. As pH decreases, ionic species, which covalently bind to casein strands, become protonated during the curd formation. This subsequently increases the hydrophobic interactions between protein molecules, making the curd harder [66]. In this regard, Ong et al. [63] reported that cheese made with milk renneted at pH 6.5 was harder than cheese made with milk at pH 6.1. This can also be true in the case of the current study, where the pH of the samples containing encapsulated SE was lower than the control cheese.

Najafi and Moatamedzadegan [7] reported that the level of fat in cheese had a direct effect on its texture, whereas the combination of animal fat in the formulation of ricotta cheese increased its adhesiveness. Nevertheless, these findings were not in agreement with those reported by Gunasekaran and Ak Gunasekaran and Ak [67], who reported that the adhesiveness of cheese was inversely related to its fat content. Ortiz et al. [36] found that low-fat ricotta cheese was harder than full-fat ricotta. However, these findings reported in the previous experiments do not apply in the case of our findings in the current study because there was no significant difference in the fat content of liposomal fortified cheeses and the control cheese (Table 3).

Lecithin is a basic and major substance in liposomes structure. In this regard, Drake et al. [59] reported that using lecithin at the concentration of 0.5% in reduced-fat cheese created a pasty texture, attributing to the interactions with the casein matrix. Considering the microstructure, these researchers observed that the casein matrix was highly disrupted, while the retention of moisture in cheese resulted in defects in its texture. Nonetheless, when it was used in lower concentrations, moisture in cheese did not interfere with its protein matrix, and thus, the typical texture of the cheese was protected. Therefore, in the case of the present study, although the presence of lecithin did not result in any change in the fat content of cheese samples, it could be one of the reasons for the increase in gumminess and chewiness of the cheese samples containing liposomal SE.

## 4. Conclusions

Taken together, the data obtained from this study showed that saffron extract could be successfully encapsulated within nanoliposomes and incorporated into a ricotta cheese product. These nanocarriers of saffron extract were stable against environmental factors and preserved saffron bioactives throughout the storage, making them suitable for the enrichment of food products such as ricotta cheese. The addition of nanoliposomes containing saffron extract, although decreased the pH of ricotta cheese to a degree, it did not significantly affect most compositional parameters of the cheese. Such fortification affected the hardness and chewiness of the control cheese with no significant effect on other textural properties. The successful manufacture of a functional ricotta cheese containing nanoliposomal encapsulated saffron extract reported in this study can be of interest for both food manufacturers and scientists, where the delivery of natural bioactives using a suitable delivery vehicle and in the most convenient way (i.e., food) is sought. Future research will focus on the antioxidant activity of the new functional ricotta cheese, the microstructure of nanoliposomes and their location in the cheese matrix, release behavior in vitro, and the effect of nanoliposomes on cheese microstructure and its microbiological properties during the storage.

## Figures and Tables

**Figure 1 molecules-27-00120-f001:**
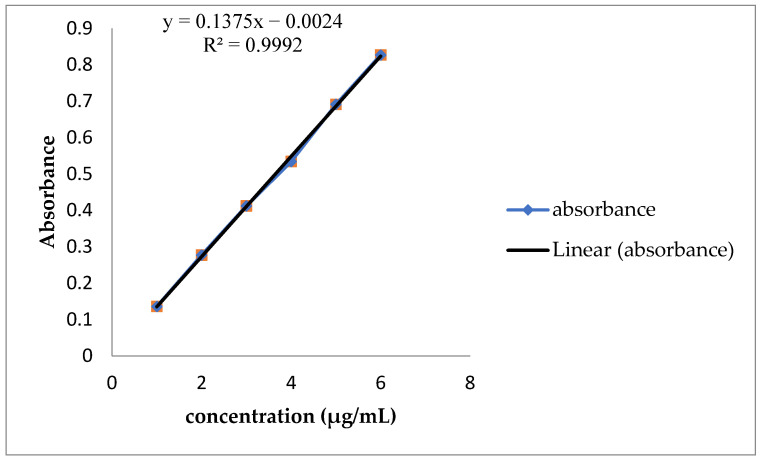
Standard curve of crocin.

**Figure 2 molecules-27-00120-f002:**
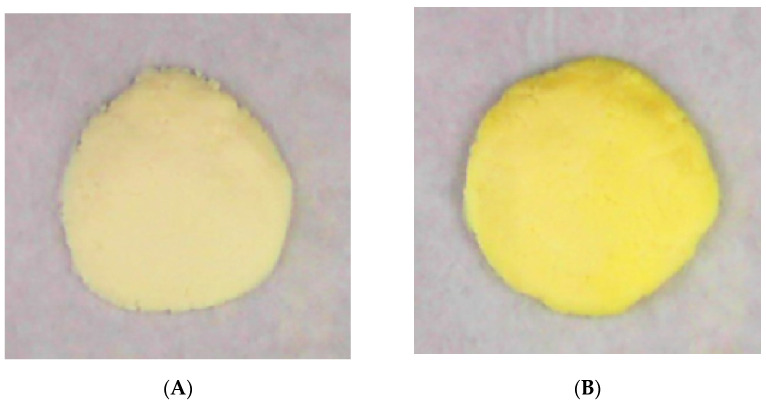
The appearance of ricotta cheese samples with no liposomes (**A**) and 2% liposomal capsulated saffron extract (**B**).

**Table 1 molecules-27-00120-t001:** Mean Z-average particle diameter, zeta potential, encapsulation efficiency, and stability of liposomes containing the saffron extract.

Type of Liposomes	Z-Average Diameter (nm)	Zeta Potential (mV)	Encapsulation Efficiency (%)	Physical Stability (%)
Liposomes containing 0.5% (*w*/*v*) lecithin	155.9 ± 1.17 ^d^	−34.6 ± 0.3 ^d^	50.73 ± 7.57 ^b^	98.54 ± 1.46 ^a^
Liposomes containing 1% (*w*/*v*) lecithin	167.8 ± 4.72 ^c^	−36.8 ± 0.8 ^c^	54.78 ± 02.93 ^b^	96.16 ± 3.83 ^ab^
Liposomes containing 2% (*w*/*v*) lecithin	190.2 ± 5.35 ^b^	−41.7 ± 0.3 ^b^	57.30 ± 1.25 ^b^	94.13 ± 3.93 ^ab^
Liposomes containing 4% (*w*/*v*) lecithin	208.1 ± 8.25 ^a^	−43.4 ± 0.3 ^a^	67.02 ± 5.93 ^a^	91.85 ± 1.09 ^b^

^a–d^ Different superscripted letters indicate significant differences among samples for each parameter (*p* ≤ 0.05) within the same column.

**Table 2 molecules-27-00120-t002:** Correlation matrix between particle size and zeta potential, encapsulation efficiency, and physical stability of liposomes containing the saffron extract.

	Zeta Potential	Encapsulation Efficiency	Physical Stability
Particle size	−0.98	0.95	−0.99

**Table 3 molecules-27-00120-t003:** The chemical composition and pH of ricotta cheese enriched with various concentrations of liposomal saffron extract.

Parameters			Treatments	
	Control (with No Liposomes)	Enriched Ricotta Cheese (mg/Kg)		
		0.125%	1%	2%
Protein (%)	17.72 ± 0.04 ^a^	17.70 ± 0.02 ^a^	17.70 ± 0.01 ^a^	17.75 ± 0.02 ^a^
Moisture (%)	53.22 ± 0.31 ^b^	53.76 ± 0.78 ^b^	54.89 ± 0.24 ^a^	55.55 ± 0.72 ^a^
Dry matter (%)	46.78 ± 0.31 ^a^	46.24 ± 0.78 ^a^	45.10 ± 0.24 ^b^	44.44 ± 0.72 ^b^
Fat (%)	19.83 ± 0.28 ^a^	19.33 ± 0.57 ^a^	19.500 ± 0.86 ^a^	19.66 ± 0.76 ^a^
Ash (%)	2.27 ± 0.69 ^a^	2.07 ± 0.3 ^a^	2.01 ± 0.25 ^a^	2.04 ± 0.07 ^a^
pH	5.63 ± 0.03 ^a^	5.62 ± 0.04 ^a^	5.41 ± 0.02 ^b^	5.39 ± 0.01 ^b^

^a,b^ Different superscripted letters indicate significant differences among samples for each parameter (*p* ≤ 0.05) within the same row.

**Table 4 molecules-27-00120-t004:** Color parameters of ricotta cheese enriched with various concentrations of liposomal saffron extract.

Parameters			Treatments	
	Control (with No Liposomes)	Enriched Ricotta Cheese (mg/Kg)		
		0.125%	1%	2%
L*	91.12 ± 1.28 ^a^	91.02 ± 0.82 ^a^	89.83 ± 0.78 ^a^	84.71 ± 0.10 ^b^
a*	−3.04 ± 0.17 ^a^	−5.05 ± 0.43 ^b^	−5.67 ± 0.42 ^b^	−6.53 ± 0.45 ^c^
b*	26.85 ± 0.96 ^c^	33.23 ± 3.33 ^b^	34.58 ± 3.50 ^b^	60.30 ± 1.62 ^a^

^a–c^ Different superscripted letters indicate significant differences among samples for each parameter (*p* ≤ 0.05) within the same row.

**Table 5 molecules-27-00120-t005:** Texture profile analysis (TPA) of ricotta cheese enriched with various concentrations of liposomal saffron extract.

Parameters			Treatments	
	Control	Enriched Ricotta Cheese		
		0.125%	1%	2%
Hardness (g)	642.34 ± 92.27 ^b^	839.04 ± 48.82 ^a^	807.85 ± 104.92 ^ab^	857.69 ± 94.61 ^a^
Adhesiveness (mJ)	4.85 ± 1.05 ^a^	5.00 ± 0.69 ^a^	5.70 ± 1.70 ^a^	4.44 ± 1.68 ^a^
Cohesiveness	0.41 ± 0.29 ^a^	0.32 ± 0.61 ^a^	0.30 ± 0.05 ^a^	0.29 ± 0.09 ^a^
Gumminess (g)	202.38 ± 71.47 ^a^	254.61 ± 44.10 ^a^	182.39 ± 22.65 ^a^	206.85 ± 59.04 ^a^
Chewiness	48.75 ± 9.92 ^b^	84.60 ± 13.05 ^a^	60.71 ± 12.37 ^ab^	60.65 ± 18.43 ^ab^

^a,b,ab^ Different superscripted letters indicate significant differences among samples for each parameter (*p* ≤ 0.05) within the same row.

## Data Availability

All data of this study are presented within this study.

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
