# Peer review of "The Effect of the Liposomal Encapsulated Saffron Extract on the Physicochemical Properties of a Functional Ricotta Cheese"

_molecules, 2021, doi:10.3390/molecules27010120_

Round 1
Reviewer 1 Report
In this study, the authors aimed to assess the feasibility of the manufacture of a functional ricotta cheese using liposomal encapsulated saffron extract and the effect on the physicochemical and textural properties of this cheese. The particle size, zeta potential, physical stability, encapsulation efficiency, chemical composition and pH of ricotta cheese, color analysis and texture analysis were analyzed. This study showed that saffron extract could be successfully encapsulated within nanoliposomes and incorporated into a ricotta cheese product. These nanocarriers of saffron extract were stable against environmental factors and preserved saffron bioactives throughout the storage, making them suitable for the enrichment of food products such as ricotta cheese. But I think this work is simply measuring and analyzing some physical characteristics, which can not reflect the advanced nature of this research method and can not meet the goal of the journal.
The following are the relevant questions in the article:
1)The authors claim that loading liposomes with SE in all concentrations of lecithin increased the average diameter of liposomes, with a significant difference (p≤0.05) being observed among different treatments. But several factors can affect particle size, such as the ratio of phospholipids to the active substance, the method of preparation of liposomes, temperature, and type of the membrane stabilizer. Why didn't the author make an in-depth study on these aspects? Similarly, for Zeta potential.
2) It is recommended to conduct correlation analysis on the data in Table 1.
3)Why does the author not use multivariate statistical analysis to study the physical characteristics at different concentrations, so as to clearly explain the influence of concentration on these quantities.
Author Response
Please see the detailed response in the attached Word file.

Reviewer 2 Report
- The authors should include more information regarding the studies that have been using nanoliposomes.
- What is the efficiency of the extraction of saffron bioactives? How do you know that the saffron bioactives were present?
- Regarding the preparation of nanoliposomes containing saffron extract, is the acetate buffer toxic?
- The average particle size and zeta potential measurements need better explanation
- Regarding physical stability assays, how was the final volume of nanoliposomes measured?
- Regarding the encapsulation studies, the standard curve needs to be included. Moreover, why was methanol used to dissolve the different concentrations of pure crocin instead of chloroform?
- Why did the addition of nanoliposomes containing saffron extract decrease the pH of ricotta cheese?
Author Response

(The authors gave the same response as above.)

Round 2
Reviewer 1 Report
The manuscipt has been revised according to the review comments. I think it can be received according to this version.
Reviewer 2 Report
The authors have addressed the reviewer's comments.